# Pluripotent Core in Bovine Embryos: A Review

**DOI:** 10.3390/ani12081010

**Published:** 2022-04-13

**Authors:** Luis Aguila, Claudia Osycka-Salut, Favian Treulen, Ricardo Felmer

**Affiliations:** 1Centre de Recherche en Reproduction et Fértilité (CRRF), Université de Montréal, Saint-Hyacinthe, QC J2S 2M2, Canada; 2Laboratory of Reproduction, Centre of Reproductive Biotechnology (CEBIOR-BIOREN), Universidad de La Frontera, Temuco 4811322, Chile; ricardo.felmer@ufrontera.cl; 3Laboratorio de Biotecnologías Reproductivas y Mejoramiento Genético Animal, Instituto de Investigaciones Biotecnológicas, Universidad Nacional de San Martín (UNSAM), Buenos Aires CP 1650, Argentina; cosycka@iib.unsam.edu.ar; 4Escuela de Tecnología Médica, Facultad de Medicina y Ciencias de la Salud, Universidad Mayor, Temuco 4801043, Chile; favian.treulen@mayor.cl; 5Department of Agricultural Sciences and Natural Resources, Faculty of Agriculture and Forestry Sciences, Universidad de La Frontera, Temuco 4811322, Chile

**Keywords:** cattle, in vitro technologies, stem cells, IVF

## Abstract

**Simple Summary:**

The fusion between sperm and oocyte results in a zygote, which is a single totipotent cell with the ability to develop into a functional organism. Totipotent cells can give rise to different specialized cell types of all lineages. Understanding the interactions between cell signaling pathways, which drive the early embryo to maintain pluripotency, is essential to establishing the optimal embryonic or stem cell culture conditions for biotechnological applications in cattle. Thus, this review summarizes the core of pluripotency genes, strategies for controlling pluripotency, and the potential applications of pluripotency in in vitro production of cattle embryos.

**Abstract:**

Early development in mammals is characterized by the ability of each cell to produce a complete organism plus the extraembryonic, or placental, cells, defined as pluripotency. During subsequent development, pluripotency is lost, and cells begin to differentiate to a particular cell fate. This review summarizes the current knowledge of pluripotency features of bovine embryos cultured in vitro, focusing on the core of pluripotency genes (*OCT4*, *NANOG*, *SOX2*, and *CDX2*), and main chemical strategies for controlling pluripotent networks during early development. Finally, we discuss the applicability of manipulating pluripotency during the morula to blastocyst transition in cattle species.

## 1. Introduction

The fusion between sperm and oocyte (two highly differentiated cells) results in a zygote, which is a single totipotent cell with the ability to develop into a functional organism. Totipotent cells can give rise to different specialized cell types of all lineages [1]. Accordingly, by the early blastocyst stage, mammalian embryos are characterized by two morphologically distinct cell populations, outer and inner cells in the morula, which form the surrounding trophectoderm (TE) and the inner cell mass (ICM), respectively, during blastulation. The ICM is formed by totipotent embryonic stem cells (ESCs) that later (in a second wave) differentiate into the pluripotent epiblast (EPI, the nascent embryo proper) and the primitive endoderm (PrE), in nonrodent mammals identified as the hypoblast (HP). The TE, on the other hand, is the precursor to the placenta and the first component of the extraembryonic structures [2,3]. In bovines, many genes show differences in expression in the ICM and TE from those in mouse or human species [4]. The lineage specification in cattle seems to be directed by a different set of regulatory factors [5]. Moreover, the precise molecular interactions governing the ICM/TE specification in this species has not been totally clarified yet [6]. Thus, understanding the interactions between cell signaling pathways, which drive the ICM to maintain pluripotency, is essential to establishing the optimal embryonic or stem cell culture conditions for biotechnological applications [4]. For instance, in the mouse species, the triad of genes *OCT4*, *NANOG* and *SOX2* are essential factors underlying pluripotency in both ICM and ES cells [7,8,9]. Caudal-type homeodomain transcription factor 2 (*CDX2*) is essential for segregation of the ICM and TE lineages at the blastocyst stage by ensuring repression of *OCT4* and *NANOG* in the TE [10]. In cattle, the lineages become fully segregated at the late blastocyst stage [5], where *OCT4*, *NANOG*, and *SOX2* are also critical transcription factors related to pluripotency maintenance in the ICM. Together with *CDX2*, they are essential for early development and gene expression involved in differentiation of the ICM and TE lineages [11].

Thus, this article focuses on two main topics restricted to bovine species. We review the literature describing pluripotency features of bovine embryos cultured in vitro, focusing on the core of pluripotency genes (*OCT4*, *NANOG*, *SOX2*, and *CDX2*), and the main chemical strategies for maintaining a pluripotent state during early development. Finally, we discuss the applicability of inducing or maintaining pluripotency in vitro.

## 2. Pluripotent Core in Early Bovine Development

### 2.1. OCT4 (POU5F1)

The octamer-binding transcription factor 4 (*OCT4*) is a transcription factor that belongs to the POU transcription family domain (POU5F1), which is expressed predominantly in pluripotent cells [12].

*OCT4* contributes to maintaining cells in an undifferentiated state by modulating expression of different loci involved in pluripotency and cellular differentiation [13]. Furthermore, *OCT4* acts as a regulator of cell lineage specification beyond the morula stage and is necessary for pluripotency maintenance and *NANOG* expression [14]. For instance, silencing of *IFNT* involves quenching of the transactivation site to inhibit differentiation towards the trophectoderm [15]. The disruption of the *OCT4* gene affects blastulation but not the ability of embryos to progress up to the morula stage, suggesting that *OCT4* is not required for cell proliferation after EGA [11,16]. Although the absence of *OCT4* expression decreases the number of ICM cells and embryonic quality [17], other authors have observed that embryos with different developmental potential, such as those produced by somatic cell nuclear transfer (SCNT), in vivo-derived, and IVF embryos have similar levels of *OCT4* expression [18]. In the same line, parthenogenetic embryos having reduced expression of *OCT4* showed no reduced cell counts, suggesting that a reduction in *OCT4* expression could be not always limitative of the ICM’s viability [19]. Thus, *OCT4* transcripts are indicative of pluripotency but would not be considered as a specific marker for embryo quality.

In bovines, *OCT4* acts as a regulator of caudal-type homeodomain protein (*CDX2*) expression and trophectoderm specification [14,16], as well as of the transition of polar and mural trophoblast development [20]. Interestingly, *OCT4* is expressed in all cell embryos throughout the morula stage but then becomes restricted to cells of the ICM in the blastocyst stage in mice [1]. Initial studies found that bovine blastocysts expressed *OCT4* mRNA only in the ICM and that its presence in the TE would be the cause of high stability of the protein or due to a delay in its clearance [21]. However, other authors have found that its expression is not restricted to pluripotent cells and that it can therefore be found in both ICM and TE cells [22,23,24,25].

During development, the maternal-derived *OCT4* transcript is present in the bovine oocyte, but after fertilization, a decrease in its abundance is observed until the time of embryonic genome activation (EGA), followed by a significant increase after the morula stage [16,21,22,26,27]. In *OCT4*-KO morulae (day 5), ~70% of the nuclei were *OCT4* positive, indicating that maternal transcripts could partially maintain *OCT4* expression during early development [14]. At the morula stage, *OCT4* and *CDX2* proteins show global nuclear localization [22,28]. Interestingly, the presence of *OCT-4* and homeobox protein *NANOG* (*NANOG*) in the TE does not interfere with the expression of trophoblast-specific genes such as *CDX2* or interferon tau (*IFN-T*) [20]. Moreover, it is not possible to increase *OCT4* expression by decreasing *CDX2*, indicating that unlike that in mice, bovine *OCT4* is not regulated by *CDX2* [14,29]. One author speculated that bovine TE was regulated by different factors and/or that the regulatory region of the *OCT4* gene showed variations among species [29]. Additionally, the presence of *OCT4* in the TE is related to the maintenance of pluripotency of this tissue to conserve the plasticity of a “non-differentiating trophoblast” [20].

Given that the process of ICM-specific allocation is gradual, *OCT4* can be found in both the ICM and in surrounding TE cells of early blastocysts (7 days postfertilization (dpf)). At 8–9 dpf, *OCT4*-positive blastomeres are predominantly located in the ICM. However, they are still detected within the TE cell population [22,28], where expression levels of *OCT4* and *CDX2* do not differ between the ICM and the TE [4]. After blastocyst hatching or 9 dpf, *OCT4* is located exclusively in ICM cells [26,28,30].

### 2.2. NANOG (Homeobox Protein NANOG)

*NANOG* is a member of the homeobox family of DNA-binding transcription factors that is known to maintain the pluripotency of ESCs [8]. Functionally, *NANOG* is not required for proper segregation of the TE and ICM, but it is required for deriving and maintaining the pluripotent epiblast and for the second lineage commitment [31,32]. Moreover, it seems to be implicated in cell proliferation, probably depending on *FGF4* signaling (which is also involved in fate decision and patterning events in the early embryo) from EPI precursor cells [31]. Thus, disruption of the *NANOG* gene did not affect the blastocyst rate but resulted in a reduced total cell number [31] and an ICM composed mostly of hypoblast cells [32]. In the nascent epiblast, *NANOG* mediated repression of hypoblast markers, such as SOX17. SOX17 is dependent on MEK signaling, but its *FGF4*-induced expression depends on *NANOG*. Therefore, the establishment of the hypoblast lineage depends on epiblast-mediated FGF/MEK signaling [31]. In relation to other markers, the absence of *NANOG* resulted in lower expression of the epiblast cell marker *SOX2* and the hypoblast marker *GATA6* without affecting the trophectoderm [32]. Moreover, in bovines, the activation of *NANOG* might be *OCT4* related. Simmet et al. [14] showed that although *OCT4*-KO bovine blastocysts expressed *NANOG* at the morula stage (probably remains of maternal origin), it was depleted in later stages, suggesting that *NANOG* expression is mutually regulated with *OCT4* [14] (Figure 1).

In the bovine embryo, *NANOG* expression begins at the 8–16-cell stage, which is the time of major EGA [22,28]. Although transcripts for *NANOG* are detected in 5-cell and 8–16-cell stage embryos, the protein is not detectable at these stages [33]. Instead, it appears de novo at the morula stage and in cavitating embryos as a product of the embryonic genome, mainly from the nascent ICM [22,28,33]. In early blastocysts (7 dpi), *NANOG* is also located in both the ICM and in the surrounding TE cells [34,35]. After hatching, *NANOG* becomes exclusively ICM-specific [5,26,28,33,36], which has been confirmed by RNA sequencing approaches [4,26,37]. Within the group of cells expressing *OCT4* and *NANOG*, there are cells expressing only *NANOG* and others expressing both factors, with the *NANOG* protein being predominantly nuclear and *OCT4* nuclear and cytoplasmic [28].

### 2.3. SOX2

SRY (sex determining region Y)-box 2, also known as *SOX2*, is a transcription factor essential to maintaining self-renewal, and pluripotency and has been reported as highly expressed in bovine ESCs [38]. *SOX2* is necessary for maintaining the undifferentiated state of the bovine ICM [37]. Lacking *SOX2* resulted in a blastocyst with a reduced number of blastomeres associated with poor embryonic quality [39,40,41], indicating a role for *SOX2* in cell proliferation. Similarly, the knockdown of *SOX2* led to the formation of a blastocyst with reduced expression of *NANOG*; since absence of *NANOG* results in lower expression of *SOX2,* this suggests a mutual regulation between *SOX2* and *NANOG* [32,42,43] (Figure 1).

During early development, *SOX2* is present in the germinal vesicle and metaphase II (MII) oocyte stages, and it can persist in nuclear and cytoplasmic compartments of four- and eight-cell embryos [22,44,45]. Expression of *SOX2* in all nuclei continues in both human and cow embryos up to the formation of an early blastocyst [46]. At the eight-cell stage, it is co-expressed with *NANOG*, but at the blastocyst stage, it overlaps with *NANOG* and GATA6 in the ICM [5,26,44]. Although it is restricted to the ICM [4,38], as recently confirmed by RNA sequencing approaches [4,26,37], some embryos can also show weak *SOX2* expression in TE cells [5,26,44]. This weak presence of *SOX2* in the bovine trophoblast could also imply a delayed commitment of TE cells to differentiation [20,29].

Remarkably, disruption of *OCT4* does not affect expression of *SOX2*, suggesting that initiation of ICM formation is *OCT4*-independent [16]. In addition, unbalanced overexpression of *SOX2* has negative effects on the control of embryonic developmental potential [47]. Dysregulated expression of *OCT4* and *SOX2* in cloned blastocysts has been related to low developmental competence in cattle [48,49,50]. Thus, *SOX2* plays a key role in the formation, maintenance, and plasticity of the ICM compartment and, therefore, on embryonic quality.

### 2.4. Homeobox Protein CDX2

*CDX2* is the master regulator of TE lineage specification [16,29,33,42,51,52]. *CDX2* is a key regulator/inducer for formation and functional maintenance of TE. At the genetic level, *CDX2* regulates multiple trophoblast genes, such as *IFNT, HAND1, ASCL2, SOX15*, and *ELF5* [26,52], and it is important for maintaining the integrity and proliferation of the trophoblast tissue [11]. It is expressed during the whole period of blastocyst development and localized in the TE and ICM of bovine embryos. The *CDX2* transcript is present in oocytes, but it decreases gradually after fertilization [51]. *CDX2* protein is found in the cytoplasm of all cells of five-cell embryos, but at the subsequent stages, it is found in the cell nuclei [28]. *CDX2* is present at the time of major EGA (8–16-cell stage) and increases afterward from the morula to the blastocyst stage [16,28,29,33,42,52]. At 7–8 dpf, *CDX2* segregation to the trophoblast cells can be noted; however, a weak signal is still present within the ICM cells during the time from expanded to hatched blastocysts [28,29,51]. At more advanced developmental stages after 9 dpf, the level of *CDX2* is at least three times higher in the TE than in the ICM [28]. In particular, *CDX2* transcripts start exceeding those of *OCT4* in the TE after hatching around day 9 [29]. Moreover, *OCT4* is not required to suppress *CDX2* in the bovine ICM [14]. Although it has been reported that *CDX2* overexpression downregulates *OCT4* [52], others have observed that *OCT-4* expression is unaffected by *CDX2* downregulation and that the deletion of the *OCT4* gene does not affect *CDX2* expression in the bovine TE [14,42], ruling out a mutual regulation (Figure 1).

In mice, specification of the TE lineage from the pluripotent early blastomeres involves the Hippo signaling pathway, with activation of *CDX2* and *TEAD4* (another transcription factor) playing a decisive role [53,54]. Similarly, the *TEAD4* transcript is present at the morula stage in the bovine embryo [29], which would activate *CDX2* to establish TE lineage [5]. A recent study confirmed that the ICM of bovine possesses the potency to become TE through the YAP1–TEAD4 axis [55]. Thus, although TE cells of the late expanded blastocyst are prone to remaining trophectoderm, they are not yet committed to this fate [29,55].

Interestingly, *CDX2*-knockdown (*CDX2*-KD) bovine blastocysts form normal blastocoel cavities and cell numbers and allocations and hatched normally without affecting *OCT4*, *NANOG*, or *SOX2* [51]. Moreover, the absence of *CDX2* promotes the overexpression of *TEAD4*, probably as a compensatory mechanism. Therefore, expression of *TEAD4* may contribute to regulating bovine blastocyst formation along with *CDX2* [51].

## 3. Chemical Modulation of Pluripotency in Early Bovine Development

### 3.1. WNT (Wingless-Related Mouse Mammary Tumor Virus) Pathway

The WNT signaling pathway is a well-known evolutionary and conserved pathway that regulates crucial aspects of cell fate determination and embryonic development [56]. In cattle, there have been several studies reporting contrasting effects of the activation/inhibition of WNT signaling during the early period of embryonic development (Figure 2, Table 1). One study showed that the activation of WNT signaling by blocking glycogen synthase kinase (GSK3) activity with LiCl2 or CT99021 had inconsistent effects on development to the blastocyst stage. LiCl decreased the proportion of zygotes reaching the blastocyst stage, while CT99021 increased this proportion [57]. Later, a study by Kuijk et al. [33] showed that embryos treated from the zygote to the blastocyst stage in the presence of the GSK3 inhibitor CHIR99021 at 3 µM had higher percentages of *NANOG* cells in the ICM. In addition, when the GSK3 inhibitor was present from the morula stage onwards, they saw no effects on ICM constitution [33]. Denicol et al. [40] observed that activation of canonical WNT signaling with the agonist AMBMP from day 5, disturbed development until the blastocyst stage and reduced the numbers of TE and ICM cells. This was not surprising, since this molecule also disrupts microtubule organization [58]. Another study observed that blocking GSK3 with CHIR99021 (3 µM) from the morula stage onwards improved blastocyst morphology and epiblast-specific gene expression (*NANOG*, *SOX2*) [59]. Similarly, Madeja et al. [12] indicated that WNT activation with CHIR99021 increased the expression of *OCT4* and *NANOG* in the ICM and downregulated *CDX2* expression. Meng et al. [36] used forskolin, which activates adenylate cyclase and cAMP/PKA signaling pathway, in turn inactivating GSK3 and thus acting synergistically with WNTs. Forskolin increased *NANOG* expression threefold [36]. More recently, Warzych et al. [5] also observed that WNT signaling (activated by CHIR99021) increased the levels of *NANOG* and *OCT4* transcripts and *NANOG*-positive cells within the ICM. Furthermore, the proportion of *OCT4*-positive cells increased in the TE concomitantly with the downregulation of *CDX2* [5]. Likewise, Sidrat et al. [60] used 6 bromoindurbin-3’oxime (6-Bio) as a WNT agonist, observing a higher expression of peroxisome proliferator-activated receptor-delta (PPARδ), which colocalized with Βeta-CATENIN and formed a complex with TCF/LEF transcription factor. In addition, 6-Bio enhanced the expression of *Βeta-CATENIN*, *OCT4*, *AXIN2*, and *C-MYC*, but *CDX2* was downregulated. Moreover, the inhibition of PPARδ with Gsk3787 severely perturbed blastocyst formation and hatching, suggesting an important role for PPARδ as a candidate regulator of the canonical WNT pathway.

On the other hand, other authors have indicated that inhibition of canonical WNT signaling regulates blastocyst development and quality. For instance, Denicol et al. [66] found that the WNT antagonist DKK1 added from the morula to the blastocyst stage promoted differentiation of cells towards trophectoderm and hypoblast lineages [66]. Similarly, exposure to Wnt-C59, which blocks secretion of WNTs, or DKK1, and interferes in the activation of the WNT-FZD-LRP5/6 receptor complex, did not affect development, but Wnt-C59 increased the number of ICM cells, suggesting that regulation of ICM proliferation by endogenous WNTs is independent of the canonical signaling [41]. However, it was recently demonstrated that inhibition of canonical WNT signaling by using an IWR1 inhibitor was crucial for ICM proliferation and derivation of bovine ESCs [38]. Another study indicated that the WNT inhibitor IWP2 increased the total cell number in blastocysts by increasing the number of TE cells and the number of *NANOG*-positive cells within the ICM but decreasing the percentage of blastocysts [44]. The differences among previous studies could be due to the different specificities and efficacies of the WNT inhibitors used [44]. In fact, recently, Xiao et al. [39] evaluated the effects of different WNT inhibitors on the derivation efficiency of bovine ESCs. They found that canonical WNT signaling was antagonist to pluripotency and that derivation of pluripotent bovine ESCs involved the inhibition of WNT signaling.

However, not all inhibitors showed the same efficacy, with IWR-1 and IWP2 being effective, unlike XAV939 and DKK1. In addition, it was observed that IWR1-inhibition between days 4 and 7.5 after fertilization blocked activation and differentiation into a pSTAT3 positive cell lineage. In the mouse embryo, Stat3 induces differentiation towards the TE lineage when its activation level exceeds certain thresholds [67]. Furthermore, CHIR99021 depressed expression of both *NANOG* and *SOX2* in bovine ESCs and decreased the number and percentage of blastomeres positive for *NANOG* and *SOX2* in the embryo [39]. In this line, other studies have indicated that TE cells highly express transcripts related to WNT signaling [30,37], as observed in the activation of WNT signaling enabling the derivation of the trophoblast stem cell by regulating *CDX2* expression through the WNT-YAP/TAZ signaling pathway [68].

Overall, the data indicate that the effects of WNT activation/inhibition depend on the specificity of the inhibitor and time of exposure. In addition, WNT signaling plays a role in TE specification, and the use of specific inhibitors able to interact with JAK and WNT signaling pathways enables the induction of epiblast pluripotency in the blastocyst.

### 3.2. MEK/ERK Pathway

Molecular interactions of signaling pathways such as MEK/ERK and WNT/β-catenin are critical for cell-to-cell communication and cellular differentiation. Secreted uterine FGF factors induce lineage commitment by activating the mitogen-activated protein kinase (MAPK), comprising MAPK kinase 2 (MAP2K, also known as MAPKK or MEK) and MAPK1/2 (ERK). It has been reported that *FGF4* mRNA is present in the trophectoderm of spherical bovine blastocysts [20]. *FGF4* can induce the formation of hypoblast and block the formation of epiblast precursors [33], but the role of *FGF4* in bovine embryo development differs from that in mice, since *FGF4* and MAPK signaling is not essential for bovine hypoblast specification [33].

The suppression of MEK signaling by PD98059 or PD325901 has been performed in numerous studies to detect the importance of MEK/ERK signaling in early development in bovines, although with controversial outcomes. Inhibition of MEK in bovine embryos resulted in ICM with increased epiblast precursors (*NANOG*+) and decreased hypoblast precursor (GATA6) [33]. Blocking bovine MEK with PD0325901 (0.4 μM) was correlated with improvement in blastocyst morphology and increases in epiblast-specific gene expression (*NANOG*, *SOX2*) [59,62]. In addition, trophoblast proliferation, lineage specification, and blastocyst formation were not affected [59,62,69]. This was consistent with studies showing that isolated trophoblast cells did not require active FGF/MEK signaling to survive, proliferate, and maintain *CDX2* expression [62,70,71,72]. In agreement with Kuijk et al. [33], under MEK inhibition (PD0325901 0.5 and 2.5 μM), neither embryonic development nor cell numbers were affected, but the proportion of *NANOG*-positive cells was markedly increased, while the expression of GATA6 was reduced but not completely switched off [31].

The effects of MEK inhibition seem to be dose dependent [5,44]. A study by Canizo et al. [44] indicated that MEK inhibition did not promote epiblast fate but rather prevented hypoblast segregation in cattle. MEK inhibition with PD0325901 at 0.4 μM decreased the numbers of ICM cells, but it had a trophic effect on the TE. Instead, high concentrations of MEK inhibition (between 1 and 2 μM) resulted in abolition of hypoblast segregation, and 10 μM affected both the TE and ICM compartments [44]. Similarly, another study indicated that MEK/ERK downregulation (PD0325901, 1 µM) maintained *OCT4* and *NANOG* within the ICM and prevented their exclusion from the TE, but *CDX2* was downregulated [5].

### 3.3. The Use of Two Inhibitors (2i) and Three Inhibitors (3i) in Early Bovine Embryonic Development

Both the 2i and 3i systems operate within the WNT and the MEK/ERK signaling pathways but use a different set of inhibitors. The 2i system (referring to the combined use of two inhibitors) includes CHIR99021 and MEK inhibition (PD0325901). The 3i (three-inhibitor) system is based on the use of 2i by a MEK/ERK inhibitor (PD184352) and GSK3 inhibitor (CHIR99021) plus an FGF receptor inhibitor (SU5402) [61]. Thus, the 2i system plus the FGF receptor inhibitor (SU5402) involves the suppression of the MAPK/ERK pathway, whereas the inhibition of GSK3 supports WNT activity.

Early studies found that the double inhibition (2i) of MEK and GSK3 offered defined culture conditions for blocking exit from pluripotency. The use of 2i enhanced bovine blastocyst development and expression of epiblast *NANOG* and *SOX2* markers by reducing expression of the hypoblast marker *GATA4* [59]. The presence of 2i (0.4–10 μM) from the morula stage (D5) onward increased the numbers of ICM cells, but *NANOG* and *FGF4* were upregulated, and specification towards the hypoblast was reduced, in the ICM after exposure to 3i combinations [62]. From day 2 onward, 3i improved embryonic development-affecting ICM-related genes (*OCT4*, *SOX2*, and *NANOG*) [34]. However, other authors found positive effects of 2i only on blastocyst quality according to total cell and ICM number [61]. Similarly, Warzych et al. [5] observed higher levels of epiblast-related genes (*NANOG* and *OCT4*) under the 2i system but no effect on the number of cells in the blastocyst. Likewise, Kuijk et al. [33] did not find any synergetic effects between CHIR99021 and PD032590, as it was recently indicated that modulation of WNT is not sufficient to support enhanced *NANOG* expression in the epiblast when combined with the ERK inhibitor [44] (Figure 2, Table 1).

Additionally, these pathways seem to be involved in the regulation of apoptosis. A study by Madeja et al. [61] found positive effects of 2i on the ICM constitution; however, the total cell counts in 3i-cultured embryos were reduced. Embryos cultured under 2i or 3i systems also showed higher rates of apoptosis and lower embryonic quality but without changes in *BAX, BCL2*, or *BAK* transcripts, suggesting alternative pathways involved in this apoptotic activation.

### 3.4. JAK/STAT

The Janus kinase/signal transducers and activators of transcription (JAK/STAT) signaling pathway mediates cellular responses to growth factors (e.g., EGF) and cytokines (e.g., IL-6). These responses include differentiation, proliferation, apoptosis, migration, and cell survival, depending on the cellular context. Thus, this signaling pathway is essential for numerous homeostatic and developmental processes, including stem cell maintenance [73].

In a study by Meng et al. [36], the authors observed that chemically suppressing JAK/STAT signaling (via JAK2/3) with AG490 and JAK1/2 inhibition with AZD1480 strongly compromised blastocyst development and quality and ICM numbers without affecting the TE. In addition, *NANOG* was reduced under both AG490 and AZD1480 treatments. The latter also strongly reduced *SOX2, KFL4*, *FGF4*, and hypoblast markers (*SOX17, PDGFRα*) without affecting *CDX2*. In addition, phosphorylation of STAT3 tyrosine (Y) 705, which is related to JAK1 pluripotency-signal [74], colocalized with *NANOG* and *SOX2* within the ICM in D7 and D8 blastocysts [36], suggesting its role in ICM specification (Figure 2, Table 1).

On the other hand, locally secreted *FGF4* can activate both (i) mitogen-activated protein kinase (MAP2K) and (ii) phosphatidylinositol 3-kinase (PI3K)–AKT [5,36,75]. JAK/STAT activation is also triggered by leukemia inhibitory factor (LIF) and related members of the interleukin (IL) family. In bovines, LIF added to the culture medium from four-cell yielded no significant benefit [76]. However, when added to the culture medium from days 5 to 8, it showed adverse effects on in vitro embryonic development based on kinetics, morphology, cell count, and the expression of *OCT4* [63]. Similarly, other authors have indicated that LIF did not affect trophoblast or ICM cell numbers [64], nor were any detrimental effects on blastocyst development observed that might be the result of the antimitotic effect of LIF, especially when used in early cleavage stages [65]. Recently, Canizo et al. [44] reported that LIF added to a 2i cocktail boosted the blastocyst yields, and LIF alone promoted expansion of hypoblast in bovine embryos, suggesting that LIF has embryotropic effects in the ICM by increasing *NANOG* and SOX17 markers. Thus, JAK/STAT signals are required for bovine ICM formation and acquisition of pluripotency markers.

## 4. The Role of Pluripotency in Biotechnological Applications

### 4.1. In Vitro Embryo Production

IVP technology has become commercially viable and extensively used for producing embryos in cattle [77]. It is also known that in vitro culture conditions determine embryo quality, expressed as developmental kinetics, blastomere count, EGA efficiency, gene expression, apoptotic rates, etc. [78]. Thus, modifications of the culture system, particularly before the time of EGA, can significantly impact the pluripotency profile and quality of the resulting blastocysts. For example, activation or inhibition of the WNT and silencing of the MEK/ERK signaling pathways alters critical pathways associated with apoptosis, implantation, and maternal recognition of pregnancy [39,61]. However, only few studies have evaluated if control of pluripotency at preimplantation stages can influence postimplantation, delivery, and/or in vivo development in bovine species. For instance, a study by Tribulo et al. [79] found that calves derived from embryos exposed to DKK1 from the morula to the blastocyst stage had lower birth weights than the control group, suggesting that changes in molecular signaling during early developmental stages impact the postnatal phenotype. Recently, Han et al. [34] evaluated the developmental effects of a modified 3i system on bovine and mouse IVF efficiency, and they transferred mouse 3i embryos to surrogate females. They did not find any differences in birth rate, sex ratio, morphology, or body weight compared with the progeny of the control group. In addition, the 3i offspring produced normal pups, indicating that the fertility of mice developed from the inhibited embryos was not affected. In this sense, it would be important to continue studying physiological changes induced by chemical inhibitors to gain greater insight into later impacts on pre- and postimplantation development.

It is well known that most embryos generated by IVF technologies (IVF, SCNT, or ICSI) do not gather the required morphological quality to be transferred [48]. For instance, bovine embryos with low development potential show a precarious balance between pluripotency factors that disturbs later stages of embryonic development [22]. Therefore, the chemical control of cell differentiation pathways and pluripotent profiles raises as a valid strategy for “rescuing” the developmental potential of embryos of lower quality to obtain embryos in vitro efficiently, especially in large animals (Figure 3).

In addition, another approach used to optimize in vitro embryo production efficiency in cattle species has been to supplement culture media with biologically active molecules produced by the reproductive tract or embryo in early pregnancy. Hansen et al. called these “embryokine”, such as CSF2, molecules produced by the female reproductive and embryo tract that control embryonic development and pluripotency [80]. This topic has been reviewed in detail elsewhere [80,81]. Thus, the control of molecular interactions of signaling pathways critical for cellular differentiation and pluripotency leads to strategies seeking to optimize IVP conditions and boost embryonic developmental potential.

### 4.2. Capturing Pluripotency In Vitro

Currently, because of an improved understanding of pluripotency, stemness from cattle species can be captured in vitro [82] by deriving embryonic stem cells from biparental embryos produced by IVF. Pluripotent stem cells can also be derived from somatic cells, the induced pluripotent stem cells (iPSCs) [83,84]. Bogliotti et al. [38] employed fibroblast growth factor 2 (FGF2) and a canonical WNT signaling pathway inhibitor in their culture conditions and derived stable pluripotent cell lines from bovine blastocysts. Bovine pluripotent cells express the pluripotent markers *SOX2* and POU5F1 and are negative for *CDX2* and the hypoblast marker GATA6 [82,85,86]. Thus, inhibition of WNT signaling by IWR-1 and stimulation of the FGF2 pathway seem to be essential requirements for deriving bovine ESC lines [86]. Xiao et al. [39] evaluated the effects of different WNT inhibitors on the derivation efficiency of bovine ESCs. They found that canonical WNT signaling was antagonistic to pluripotency and that derivation of pluripotent ESCs involved inhibition of WNT signaling. Nonetheless, not all inhibitors showed the same efficacy, with IWR-1 and IWP2 being effective but not XAV939 and DKK1. Recently, Soto et al. [86] reported a simplified bESC culture system based on a commercially available medium (N2B27) and feeder-free culture conditions based in a chemical substrate (Vitronectin) supplemented with activin A (AA). Activin is a growth factor known to support the expansion of human ESCs [87]. Nonetheless, the effects of AA on the developing bovine blastocyst are adverse for ICM proliferation [88], which would restrict its use only for bESC derivation.

On the other hand, bovine iPSCs (biPSCs) have been generated from somatic cells using exogenous transcriptional factors combined with small chemical inhibitors supported by current knowledge of pluripotential pathways. Using a combination of seven factors (*OCT4*, *SOX2*, *NANOG*, *KLF4*, *cMYC*, *LIN28*, and *KDM4A*) and a reprogramming medium containing inhibitors of WNT (IWR1) and H3K79 methyltransferase Dot1L (iDot1L), Su et al. [84] derived primed-like iPSCs from mesenchymal stem cells. *OCT4*, *NANOG*, and *SOX2* were highly activated in these iPSCs across different passages [84]. Similarly, Pillai et al. [83] enhanced the cellular reprogramming of bovine fibroblasts to biPSCs by forcing expression of *OCT4, SOX2, KLF4*, and *MYC*, but they also reported that inhibition of ALK4/5/7 to block TGFβ/activin/nodal signaling together with GSK3β and MEK1/2 supported robust in vitro self-renewal of naive biPSCs. A detailed review of these topics was recently reported [89].

## 5. Future Perspectives

In general, chemical approaches have been quite successful in modulating and identifying pluripotency pathways involved in early development and cell fate differentiation of mammalian embryos. However, off-target effects complicate small-molecule methods. For example, a study by Xiao et al. [39] found that derivation of bovine ESC can be conducted by IWR-1 and IWP2 but not by XAV939 and DKK1, either because they do not specifically inhibit WNT signaling or because they have additional effects that affect cell function in a canonical, WNT-independent manner. One alternative tool for investigating the functional genomics of bovine pluripotency is gene editing approaches. For instance, Daigneault et al. [16] used CRISPR/Cas9 for targeted disruption of the POU5F1 (*OCT4*) gene by direct injection into zygotes. Disruption of the bovine POU5F1 locus was highly efficient and was associated with developmental arrest at the morula stage, indicating that POU5F1 is essential for the formation of expanded bovine blastocysts. Similar approaches have been reported [32,90,91]. Likewise, RNA interference has greatly facilitated analysis of loss-of-function phenotypes [92], but correlating these phenotypes with small-molecule inhibition profiles is not always straightforward [93]. Jafarpour et al. [94] reported that downregulation of SUV39H1/H2 (a histone methyltransferase) through siRNA in fibroblasts improved the blastocyst yield of bovine SCNT embryos. However, this effect was not observed after the treatment of fibroblasts with chaetocin, a chemical inhibitor of SUV39H1/H2, possibly because of its off-target effects on other histone methyltransferases. Indeed, gene editing technologies have a wide range of applications in livestock species [95,96], but they also represent valuable tools for investigating functional genomics of the bovine embryo at early stages.

Finally, it is important to highlight the contribution of high-throughput sequencing platforms for dissecting pathways and identifying undefined gene expression sequences associated with pluripotency [83,97,98]. Thus, transcriptomic analysis will continue contributing to the molecular characterization of bovine pluripotency and to the establishment of culture conditions that support the derivation and maintenance of true ESCs.

## 6. Conclusions

In this article, we review the current knowledge of core pluripotency markers during early development of bovine embryos. We also describe the main pathways involved in pluripotency maintenance and cell differentiation. Embryonic pluripotency depends on the activation of several molecular mechanisms involving different factors. The rigorous balance between maternal clearance and zygotic expression of *OCT4, NANOG, SOX2*, and *CDX2* affects differentiation, proliferation, apoptosis, and embryonic quality. In addition, WNT signaling seems to play a crucial role in driving both cell differentiation and pluripotency maintenance in bovine species. The recent derivation of PSCs (ESCs and iPSCs) is a hallmark of progress concerning pluripotency in cattle species. There is also great potential to optimize in vitro culture conditions by controlling cell differentiation networks, such as by incorporating small molecules and avoiding the need for undefined culture components. In this vein, manipulating pluripotency networks of low-quality embryos produced by IVF technologies can certainly rescue their developmental competence and increase the efficiency of IVP, especially in large species. However, chemical modulation will depend on exposure time, concentrations, and secondary targeting of the small molecule(s). Even the base medium used (e.g., SOF, KSOM, N2B27, etc.) can undoubtedly influence the final outcome. Indeed, the generation of more specific antibodies, genetic engineering, and advanced technologies such as deep sequencing approaches has contributed significantly to understanding how early development in mammals diverges in terms of pluripotent characteristics and to establishing favorable conditions for capturing cattle pluripotency in vitro.

## Figures and Tables

**Figure 1 animals-12-01010-f001:**
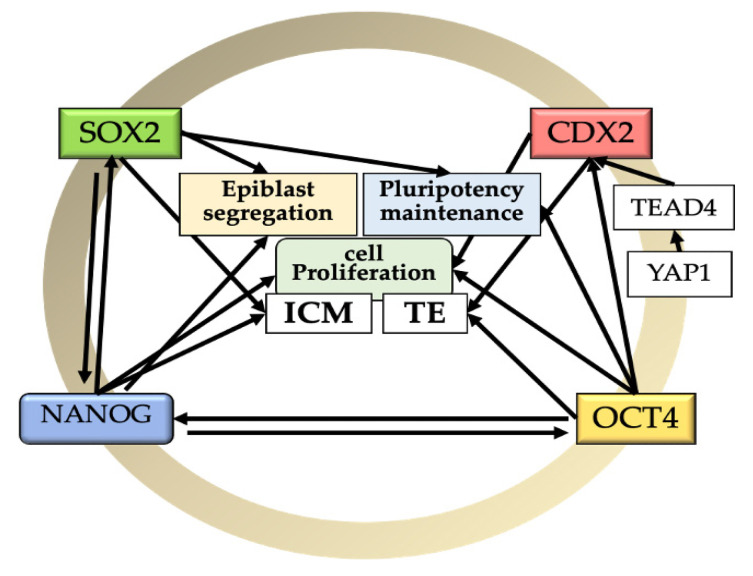
Relationships among the core of pluripotency factors and epiblast segregation, pluripotency maintenance, and cell proliferation during early bovine development. *SOX2*: SRY (sex determining region Y)-box 2; *OCT4*: octamer-binding transcription factor 4 (POU5F1); *NANOG*: homeobox protein *NANOG*; TE: trophectoderm; ICM: inner cell mass; *TEAD4*: TEA Domain Transcription Factor 4; YAP1: Yes Associated Protein 1. Black arrows indicate positive correlations.

**Figure 2 animals-12-01010-f002:**
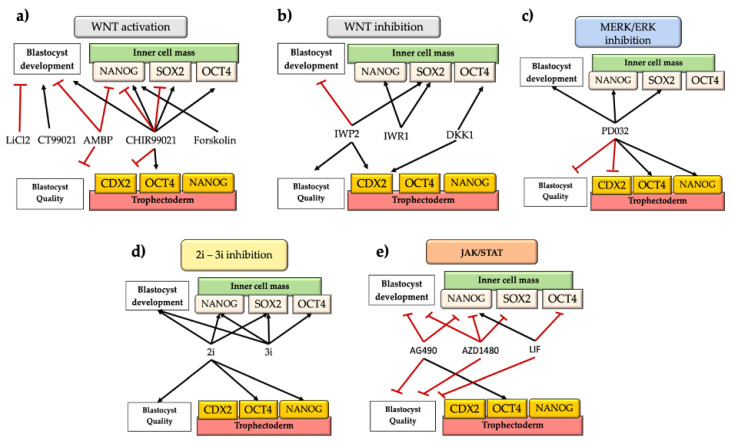
Effects of small molecules on levels of pluripotency factors (*NANOG*, *SOX2*, *OCT4*, and *CDX2*), on developmental potential in vitro and embryo quality according to total cell number. (**a**) Effects of WNT activation on in vitro development and embryo quality; (**b**) Effects of WNT inhibition on in vitro development and embryo quality; (**c**) Effects of MERK/ERK inhibition on in vitro development and embryo quality; (**d**) Effects of 2i-3i inhibition on in vitro development and embryo quality. (**e**) Effects of JAK/STAT inhibition/activation on in vitro development and embryo quality. Note that some molecules have shown opposites effects across the literature reviewed. *SOX2*: SRY (sex determining region Y)-box 2; *OCT4*: POU5F1; *NANOG*: homeobox protein *NANOG*; *CDX2*: homeobox protein *CDX2*; WNT: wingless-related mouse mammary tumor virus pathway; MEK/ERK: Ras/Raf/Mitogen-activated protein kinase/ERK kinase (MEK)/extracellular-signal-regulated kinase (ERK); 2i-3i inhibition: two-three inhibitors systems (MEK inhibition+WNT activation, and MEK inhibition+WNT activation+FGFR inhibitor, respectively); JAK/STAT: Janus kinase-signal transducer and activator of transcription. Black arrows indicate positive correlations, and red lines indicate negative correlations.

**Figure 3 animals-12-01010-f003:**
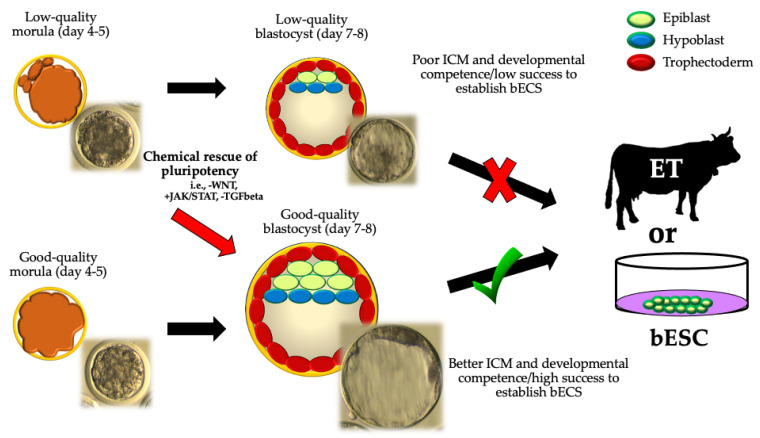
A theoretical approach to “rescue” in vitro developmental potential from cattle embryos of lower quality (seen as morula with delayed compaction, presence of cell debris, fragmentation and/or slower developmental kinetics). ICM: inner cell mass; ET: embryo transfer; bESCs: bovine embryonic stem cells). Low-quality blastocyst: embryo with delayed blastulation, poor symmetry, and/or cells that are loosely packed for the ICM and trophectoderm. Good-quality blastocyst: embryo with an expanded blastocoel cavity, highly symmetric, absence of cell debris or fragmentation, and highly packed ICM and trophectoderm cells, where a clearly visible ICM can be distinguished during morphological valuation. -WNT: WNT-inhibition; +JAK/STAT: JAK/STAT activation; -TGFbeta: TGFbeta inhibition.

**Table 1 animals-12-01010-t001:** Small-molecule inhibitors and their effects on pluripotency and development of bovine embryos.

Reference	Days in Culture	Molecule	Effect	Concentration	Effects on Pluripotency-Related Factors	Effects on Early Development
**1. WNT PATHWAY**						
[12]	day 1–9	Chir99	WNT activation	3 μM	Higher expression of *OCT4* and *NANOG* in the ICM and TE not accompanied by an increase in the number of *NANOG*-positive cells. *CDX2* expression in the TE downregulated. GATA6 significantly upregulated. Induced specification of hypoblast markers	n.e.
[33]	day 1–8	Chir99	WNT activation	3 μM	Slightly higher percentage of *NANOG*-positive cells than control embryos. Moreover, no synergistic effect between Chir99 and PD032. Induced specification of hypoblast markers	n.e.
[36]	day 5 onward	Forskolin	WNT activation	10 μM	Increased *NANOG* expression threefold without significantly altering *FGF4* or hypoblast markers	n.e.
[39]	day 4 to 7.5	IWR1	WNT inhibition	2.5 μM	Decreased lineage of STAT3 cells. Did not affect number of cells	n.e.
[40]	day 5 onward	AMBMP	WNT activation	0.7–2.8 μM	Reduced cell numbers in the TE and ICM	Affected the development until blastocyst stage
[41]	day 5 onward	AMBMP	WNT activation	0.7 μM	Induced accumulation of β-CATENIN	Reduced development to the blastocyst stage
		Wnt-C59	WNT activation	10 μM	Induced accumulation of β-CATENIN	Decreased the proportion of oocytes and cleaved embryos becoming blastocysts
		Chir99	WNT activation	5 μM	Induced accumulation of β-CATENIN	Reduced development to the blastocyst stage
		DKK1	WNT inhibition	100 ng/mL	Decreased YAP1 in TE. Did not affect number of ICM or TE cells	No effects on embryonic development
		WNT7A	WNT activation	66 ng/ml	Inhibited the PCP pathway and did not affect amounts of β-CATENIN	Increased blastocyst development
[59]	day 5 onward	Chir99	WNT activation	3 μM	Increased cell numbers in the TE and ICM	n.e.
[60]		6-Bio	WNT activation	400 nM	High expression of *PPARδ*, β-CATENIN, *OCT4*, *AXIN2*, and C-MYC. *CDX2* was downregulated. PPARδ colocalized with β-CATENIN and formed a complex with TCF/LEF transcription factor	Enhanced blastocyst formation, quality, and hatching rates
**2. MEK/ERK PATHWAY**						
[33]	day 1–8	PD98059 or PD032	MEK inhibition	25 μM	More *NANOG*-positive cells than GATA6-positive cells. Induced specification of hypoblast markers	n.e.
[31]	day 5 onward	PD032	MEK inhibition	0.5 or 2.5 μM	*NANOG*-positive cells increased, while the expression of GATA6 was reduced	No effects on embryonic development
[44]	day 5 onward	PD032	MEK inhibition	0.4–2 μM	Increased the total number of cells by increasing TE cells but reducing ICM cell number. Prevented hypoblast segregation	No effects on embryonic development
[61]	day 1–9	2i	MEK inhibition	1 μM PD032 + 3 μM Chir99	Higher apoptosis rate. No changes in BAX, BCL2, BAK, or BAX/BCL2 ratio. Expression of embryo quality genes (HSPA1A, SLC2A1) not affected. Reduced expresion of IFNT2	No effects on embryonic development
**3. TWO INHIBITORS (2i)**						
[5]	day 1–9	2i+	MEK inhibition/ WNT activation	10 μM PD032 + 3 μM Chir99	Upregulation of *NANOG* and *OCT4* in the ICM and downregulation of *CDX2*. Increases in *OCT4* and *NANOG*-positive cells within the ICM and TE. Promoted specificatoin of epiblast markers	Improved blastocyst morphology, but no changes on cell numbers
		2i	MEK inhibition/ WNT activation	1 μM PD032 + 3 μM Chir99		n.e.
[59]	day 1–8	2i	MEK inhibition/ WNT activation	0.4 μM PD032 + 3 μM Chir99	Increased expression of *NANOG* and *SOX2* and increased cell numbers in the trophoblast and ICM. Repressed putative hypoblast marker GATA4. *OCT4* and *SOX2* were not affected	Improved development. Promoted specification of epiblast markers
[62]	day 5 onward	2i	MEK inhibition/ WNT activation	0.4 μM PD032 + 3 μM Chir99	Higher expression of *NANOG* and *FGF4* and higher cell number in ICM, but reduced PDGFR alpha and SOX17 levels	No developmental changes. Promoted specification of epiblast markers
**4. THREE INHIBITORS (3i)**						
[5]		3i/3i+	MEK inhibition/WNT activation/FGFR inhibitor	0.8 μM PD184 + 3 μM Chir99 + 2 μM SU5402	Upregulation of *NANOG* and *OCT4* in the ICM and downregulation of *CDX2*. Increase of *OCT4* and *NANOG* positive cells within the ICM and TE	n.e.
[34]	day 2–8	3i	MEK inhibition/WNT activation/WDR5 inhibition	0.5 μM PD032 + 0.5 μM Chir99 + 30 μM MM102	Reduced expression of ICM-related gene (*OCT4*, *SOX2* and *NANOG*). Increases the expression of KLF4 and KLF17 and decreases the expression of the de novo DNA methyltransferase genes DNMT3L and DNMT1	Improved blastocyst development
[62]	day 5 onward	3i/3i+	MEK inhibition/WNT activation/FGFR inhibitor	10 μM PD032 + 3 μM Chir99 + 0.1 μM PD17	Increase in TE numbers only under additional FGFR inhibition. Overexpression of *NANOG* and *FGF4* was less consistent	Improved blastocyst morphology
[61]		3i	MEK inhibition/WNT activation/FGFR inhibitor	0.8 μM PD18 + 3 μM Chir99 + 2 μM SU5402	Higher expression of GJA1 and cell-to-cell interactions transcripts	Promoted cell-to-cell interactions but lowered embryonic quality
**5. JAK/STAT PATHWAY**						
[33]	day 1–8	*FGF4*	FGF activation	1 μg/mL	Reduced numbers of *NANOG*-positive cells and enhanced numbers of GATA6-positive cells. No effects on *CDX2*. From the morula stage onwards, ICMs were composed entirely of GATA6-cells. Induced specification of hypoblast markers	n.e.
[36]	day 5 onward	AZD1480.	JAK/STAT activation	10 μM	ICM formation was affected, but trophectoderm cell numbers and markers (*CDX2*) not altered. JAK inhibition repressed both epiblast and hypoblast factors	n.e.
[44]	day 5 onward	LIF	JAK/STAT activation	20 ng/mL	Trophic effect in the bovine ICM by increasing *NANOG*- and SOX17-positive cells at day 8	Boosted the blastocyst yield and had a trophic effect on the hypoblast
[63]	day 5 onward	LIF	JAK/STAT activation	1000 U/mL	Reduced cell count and expression of *OCT4*	Adverse effects on development based on kinetics and morphology
[64]	day 6 onward	LIF	JAK/STAT activation	100 ng/mL	Decreased cell counts both in terms of inner cell mass (ICM) and ICM/total cell proportions	n.e.
[65]	day 4 onward	LIF	JAK/STAT activation	100 ng/mL	Induced specification of hypoblast markers	Promoted blastocyst development only when added on day 4. Detrimental effects on development when added at day 0 and/or from day 4 to 7

ICM: inner cell mass; TE: trophectoderm; Chir99: Chir99021; PD17: PD173074; PD18: PD184325; WNT PATHWAY: Wingless-Related Mouse Mammary Tumor Virus signaling; MEK/ERK PATHWAY: Ras/Raf/Mitogen-activated protein kinase/ERK kinase (MEK)/extracellular-signal-regulated kinase (ERK); TWO INHIBITORS (2i): two inhibitors (MEK inhibition/WNT activation); THREE INHIBITORS (3i): three inhibitors (MEK inhibition/WNT activation/FGFR inhibitor); JAK/STAT: Janus kinase-signal transducer and activator of transcription. n.e.: not evaluated.

## Data Availability

Not applicable.

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
