# Peer review of "Pluripotent Core in Bovine Embryos: A Review"

_animals, 2022, doi:10.3390/ani12081010_

Round 1
Reviewer 1 Report
This article reviews genes that comprise the pluripotent core in bovine embryos, usage of chemicals for regulating pluripotent networks, and the role of pluripotency in biotechnological applications. While the information is largely there, the manuscript would benefit significantly with the following changes.
- Line 64, section 2.1. This section should begin by talking about what OCT4 does, so it would be more appropriate to move the fourth paragraph beginning on line 95 up to the beginning of this section. After an appropriate introduction, then its regulation can be discussed.
- This review is about bovine embryos. It is stated that OCT4 transcript profile differs between bovine and mouse, but seems to make the assumption that the reader will know about the transcriptional profile of Oct4 in mouse embryos. If the authors wish to compare bovine to mouse, then they should begin by describing what’s known about OCT4 in mouse embryos.
- Section 2.1, lines 80 and 81 – CDX2 and NANOG should be defined. The authors should not make the assumption that the reader already knows about these proteins.
- Line 105, SCNT should be defined.
- Section 2.2. This section suffers from the same problem as Section 2.1, in that NANOG should be introduced to the reader before discussing its regulation. What does it do? It would be better to reorganize so that the function of NANOG is clear first, and then the regulation can be discussed.
- Section 2.3, as in the first two sections, should begin by stating what the function of SOX2 is – so move the first sentence or two from the second paragraph to the first.
- Section 2.4 – lines 198-200 should be moved earlier in the section, as they state the function of CDX2.
- While the authors point out on lines 271-272 that the effects of pharmacological WNT activators and inhibitors depend on the specificity and exposure of the particular agent, there should be a more careful assessment of which inhibitors are the most specific and reliable. The authors could then focus on discussing the results of the compounds that are the most likely to shed light on the function of the target protein rather than presenting a large table composed of pharmacological WNT activators (for example) that have disparate effects on embryo development.
- Section 3.3 – “2i” and “3i” should be defined on line 308.
- Section 3.4 – Define what JAK/STAT pathway is/does before talking about its inhibition.
- The manuscript would benefit by some light editing by a native English speaker.
Author Response
R: We appreciate the reviewer´s positive comments and useful suggestions that have helped us to improve our manuscript. We have taken all of these comments and suggestions into account in the revised version of our manuscript. Also, English was revised and improved by an English native teacher.
- Line 64, section 2.1. This section should begin by talking about what OCT4 does, so it would be more appropriate to move the fourth paragraph beginning on line 95 up to the beginning of this section. After an appropriate introduction, then its regulation can be discussed.
R: Following the reviewer´s suggestion, we have made changes on Section 2.1 (see page 2, lines 66-84).
- This review is about bovine embryos. It is stated that OCT4 transcript profile differs between bovine and mouse but seems to make the assumption that the reader will know about the transcriptional profile of Oct4 in mouse embryos. If the authors wish to compare bovines to mouse, then they should begin by describing what’s known about OCT4 in mouse embryos.
R: We have taken the reviewer´s comment into account and we have added a brief description regarding the reviewed genes in mouse species in the revised version of our manuscript. See page 2, lines 53-57 and lines 87-89)
- Section 2.1, lines 80 and 81 – CDX2 and NANOG should be defined. The authors should not make the assumption that the reader already knows about these proteins.
R: We agree with the reviewer´s observation. We have defined CDX2 and NANOG in the new version of the manuscript (see page 2, lines 52-53 and page 3, lines 100 and 114).
- Line 105, SCNT should be defined.
R: We have defined SCNT in the new version of the manuscript (see page 2, line 79).
- Section 2.2. This section suffers from the same problem as Section 2.1, in that NANOG should be introduced to the reader before discussing its regulation. What does it do? It would be better to reorganize so that the function of NANOG is clear first, and then the regulation can be discussed.
R: Following the reviewer´s suggestion, we have also made changes to Section 2.2 (see page 3, lines 115-133).
- Section 2.3, as in the first two sections, should begin by stating what the function of SOX2 is – so move the first sentence or two from the second paragraph to the first.
R: We agree with the reviewer´s observation. We have moved two sentences of the second paragraph to the first in Section 2.3 in the new version of the manuscript (see pages 3 and 4, lines 147-153).
- Section 2.4 – lines 198-200 should be moved earlier in the section, as they state the function of CDX2.
R: Following the reviewer´s suggestion, we have moved lines 198-200 earlier in Section 2.4 (see page 4, lines 176-181).
- While the authors point out on lines 271-272 that the effects of pharmacological WNT activators and inhibitors depend on the specificity and exposure of the particular agent, there should be a more careful assessment of which inhibitors are the most specific and reliable. The authors could then focus on discussing the results of the compounds that are the most likely to shed light on the function of the target protein rather than presenting a large table composed of pharmacological WNT activators (for example) that have disparate effects on embryo development.
R: We appreciate the reviewer´s comments and useful suggestions that have helped us to improve our manuscript. However, for these authors it is also important to describe compounds that have shown contrasting effects, thus this information may also be useful for readers and researchers who want to use those compounds to confirm or validate, under the same or different experimental conditions, what has been previously described.
- Section 3.3 – “2i” and “3i” should be defined on line 308.
R: Following the reviewer´s suggestion, we have defined 2i and 3i in the new version of the manuscript (see page 7, line 315)
- Section 3.4 – Define what JAK/STAT pathway is/does before talking about its inhibition.
R: We agree with the reviewer´s observation. We have defined and added more information about JAK/STAT pathway in the new version of the manuscript (see page 8, lines 345-350).
- The manuscript would benefit by some light editing by a native English speaker.
R: We appreciate the reviewer´s suggestion. In this new version, English was revised and improved by an English native speaker.

Reviewer 2 Report
This review paper summarized the critical pluripotency genes and some chemical strategies to improve the developmental potency and applications on IVP cattle embryos. The manuscript was well designed and recent advances were also covered. However, there are no any discussion about what we can do next or perspective of IVP embryos using alternative methods. So, I suggest authors will discuss some future directions to improve the IVP embryos' pluripotency.
- there was a wrong with table 1, eg. the location of reference and days in culture...etc.
- Line 407, "because to" can be "due to" or "because of".
- Before the section of Conclusions (Line 432), please discuss the future directions or perspective.
Author Response
Reviewer 2:
Comments and Suggestions for Authors
This review paper summarized the critical pluripotency genes and some chemical strategies to improve the developmental potency and applications on IVP cattle embryos. The manuscript was well designed, and recent advances were also covered. However, there are no any discussion about what we can do next or perspective of IVP embryos using alternative methods. So, I suggest authors will discuss some future directions to improve the IVP embryos' pluripotency.
R: We appreciate the reviewer´s positive comments and we would like to thank the reviewer’s efforts to improve the quality of our manuscript. We have taken all of these comments and suggestions into account in the revised version of our manuscript.
- there was a wrong with table 1, eg. the location of reference and days in culture...etc.
R: We apologize for this oversight. We have now added a new Table 1 in the new version of the manuscript (see page 9 and page 10).
- Line 407, "because to" can be "due to" or "because of".
R: Following the reviewer´s suggestion, we have changed “because to” by “due to” (see page 12, line 428).
- Before the section of Conclusions (Line 432), please discuss the future directions or perspective.
R: We apologize for not including this information in our original manuscript. We have now included this in the new Section 5 (Future Perspectives) of the revised manuscript (see page 12, lines 460-477 and page 13, lines 478-487).
